# Arylcyclohexylamine Derivatives: Pharmacokinetic, Pharmacodynamic, Clinical and Forensic Aspects

**DOI:** 10.3390/ijms232415574

**Published:** 2022-12-08

**Authors:** Romain Pelletier, Brendan Le Daré, Diane Le Bouëdec, Angéline Kernalléguen, Pierre-Jean Ferron, Isabelle Morel, Thomas Gicquel

**Affiliations:** 1Univ Rennes, INSERM, INRAE, CHU Rennes, Institut NuMeCan (Nutrition, Métabolisme et Cancer), F-35000 Rennes, France; 2Centre Hospitalier Universitaire de Rennes, Laboratoire de Toxicologie Biologique et Medico-Légale, CHU Pontchaillou, 2 Rue Henri Le Guilloux, F-35000 Rennes, France; 3Centre Hospitalier Universitaire de Rennes, Service Pharmacie, F-35000 Rennes, France

**Keywords:** arylcyclohexylamines, ketamine, phencyclidine, NPS, review

## Abstract

Since the 2000s, an increasing number of new psychoactive substances (NPS) have appeared on the drug market. Arylcyclohexylamine (ACH) compounds such as ketamine, phencyclidine and eticyclidine derivatives are of particular concern, given their rapidly increasing use and the absence of detailed toxicity data. First used mainly for their pharmacological properties in anesthesia, their recreational use is increasing. ACH derivatives have an antagonistic activity against the N-methyl-D-aspartate receptor, which leads to dissociative effects (dissociation of body and mind). Synthetic ketamine derivatives produced in Asia are now arriving in Europe, where most are not listed as narcotics and are, thus, legal. These structural derivatives have pharmacokinetic and pharmacodynamic properties that are sometimes very different from ketamine. Here, we describe the pharmacology, epidemiology, chemistry and metabolism of ACH derivatives, and we review the case reports on intoxication.

## 1. Introduction

Since the 2000s, an increasing number of new psychoactive substances (NPS) have appeared on the drug market. In most cases, a single functional or chemical group constitutes the only difference between the NPS and the parent recreational drug. This change not only alters the molecule’s pharmacological properties, but also makes the use of a molecule legal, simply because the derivative is not registered on the list of narcotic products—making it a “legal high” [1,2,3]. Due to their very low cost (5 to 20 euros per gram) and ready availability (facilitated by deliveries via the Internet), NPSs are an emerging public health problem. Arylcyclohexylamine (ACH) derivatives, such as ketamine, are of particular concern, given their rapidly increasing use and the absence of detailed toxicity data.

Although ketamine is currently the best-known member of the ACH family, the latter’s history began with the lead compound phencyclidine (PCP). Clinical trials of PCP as an analgesic (Sernyl^®^) were initiated in 1958, but abandoned in 1965 due to sometimes uncontrollable “psychotic” adverse drug reactions in ~15% of the study participants [4,5]. PCP’s psychedelic properties led to a new chapter in its history as a street drug (“angel dust”) and made it the first of many synthetic drugs to appear on the market as an illicit recreational substance [6]. PCP had its heyday in 1976, with many newspaper and television reports on its recreational use [7]. Despite stricter law enforcement controls, a move to Schedule II of the Unites States’ Controlled Substances Act in January 1978 and a fall in media coverage, consumption of PCP has been increasing steadily after a slight decline in the late 1980s.

First-generation PCP derivatives appeared at street markets between the 1960s and 1990s. The first cases of non-medical PCP use were described in 1967 and 1968 in the USA (mainly in San Francisco and New York) [8]. Most of the PCP derivatives correspond to basic chemical changes. Morris and Wallach estimated that there are 14 PCP derivatives of recreational interest [9]. Over the last 10 years, only a few ACH derivatives (e.g., 3-MeO-PCP, 2-fluorodeschloroketamine (2F-DCK) and 3-OH-PCP) have been commonly used for recreational purposes. Like ketamine, these molecules can be extremely toxic and sometimes fatal (see Section 3). Ketamine was synthesized from PCP in 1962 by the pharmaceutical company Parke-Davis (Detroit, MI, USA), which was seeking to develop new ACH anesthetics with analgesic properties [10]. The first-in-man trial took place in 1964 at Michigan State Prison (Jackson, MI, USA). Despite the presence of dissociative effects (corresponding to a dissociation between body and mind) associated with short-acting anesthesia, a patent was filed in 1966, and ketamine was approved for sale by the United States Food and Drug Administration (FDA) in 1970 [11,12]. During the Vietnam War (1955–1975), ketamine was used as an anesthetic and an analgesic [13]. In 1989, the FDA’s approval of propofol led to a decrease in the use of ketamine as an anesthetic [14]. The use of ketamine was revived a few years later, following the discovery of the drug’s value in the treatment of opioid-induced hyperalgesia as well as the treatment of depression [15]. At present, ketamine is widely used in veterinary medicine, pediatrics, anesthesia and emergency medicine. Along with ketamine’s medical uses, its powerful dissociative effect and low price make it a sought-after recreational product. In Europe, ketamine has been classified as a narcotic product since the 1990s [16].

Recreational ketamine use has been mainly described in electro-alternative culture, rave parties and, more particularly, in Asian countries. The synthetic ketamine derivatives produced in Asia are now arriving in Europe, where most are not listed as narcotics and are, thus, legal. The first synthetic derivatives (eticyclidine (PCE), 2-oxo-eticyclidine, etc.) were observed in the 1960s (at the same time as ketamine), although their effects were considered to be too powerful. Furthermore, supply problems prevented the drugs from being used widely. However, the advent of the Internet (and especially the Dark Web in the 2010s) and better logistics led to a huge increase in the consumption of ketamine and its derivatives. Cases of intoxication were first described in Asia, and are becoming more frequent in Europe [17,18,19,20]. Herein, we describe the pharmacology and metabolism of ACH derivatives and review the case reports on intoxication.

## 2. Pharmacology of Ketamine and Its Synthetic Derivatives

### 2.1. Physical-Chemical Properties

The first-generation PCP derivatives keep a cyclohexane ring, in order to retain the antagonistic activity against the N-methyl-D-aspartate (NMDA) receptor and, thus, the dissociative effects. The ACH family encompasses three main subfamilies: ketamine-like molecules, phencyclidine-like (PCP-like) molecules and eticyclidine-like (PCE-like) molecules. These compounds are mainly derived through modification of the aryl ring, i.e., through the addition of an alkyl chain or substitutions of the amine group (Figure 1).

Ketamine is characterized by three rings: an aryl ring, a cyclohexyl ring, and an amine ring (Figure 1). The cyclohexane ring is usually intact because it is required for antagonism of the NMDA receptor. ACHs are chiral; the asymmetric carbon in the C2 position has two optical isomers. Ketamine is marketed as a racemate (i.e., a mixture of S(+)-ketamine, also known as esketamine and R(–)-ketamine, also known as arketamine) or as the pure S enantiomer. Ketamine is the only ACH to have a halogen (chlorine) on the aryl ring. In 2-F-DCK, the chlorine is replaced by fluorine. In other ACH derivatives, hydroxy (OH) or methoxy (MeO) groups are added to the aryl ring [21] (Figure 1).

### 2.2. Pharmadynamics

#### 2.2.1. Mechanism of Action

ACH derivatives act mainly by antagonizing the glutamate NMDA receptor in the brain and spinal cord [22]. The NMDA receptor is an ion channel receptor consisting of a combination of four subunits encoded by seven genes: GluN1, GluN2_A→D_ and GluN3_A→B_ [23]. The channel is permeable to calcium ions, and triggers many intracellular pathways [24]. ACH derivatives bind to an allosteric site (the PCP-binding site) in the channel and act as non-competitive antagonists [25]. This antagonism is responsible for the dissociative anesthetic and amnesic effects, and (perhaps) the antidepressant and analgesic effects (Figure 2).

An ACH’s three-dimensional structure is important for its antagonistic action. Compared with R(–)-ketamine, S(+)-ketamine has four times the activity and four times the affinity for the PCP site of the NMDA receptor. Hence, S(+)-ketamine is twice as active as the racemate [26].

Although data on the pharmacology of the new synthetic ACHs are scarce, all the derivatives (including PCP and ketamine) bind to the PCP binding site [9,27] (Figure 2).

Although it is thought that ACH derivatives bind to other receptors (e.g., the muscarinic and opioid receptors, for ketamine), there are no data on relevant clinical effects [28,29,30].

#### 2.2.2. Clinical Effects

Ketamine has a powerful, rapid, anesthetic and dissociative effect, but (unlike opioids) does not cause hypotension or a respiratory depressant effect [6]. Domino et al. (1965) defined dissociative syndrome as the failure of cortical sensory information to reach associative areas, i.e., dissociation between the thalamocortical and limbic systems.

As is the case for the ACH derivatives’ anesthetic effects, the analgesic uses are limited by the compound’s psychodysleptic effects. Ketamine has several mechanisms of action, resulting in (i) a dissociative effect via the decoupling of nociceptive information from pain, (ii) inhibition of descending inhibitory systems and astrocytic pronociceptive systems, (iii) a local anesthetic effect and (iv) anti-inflammatory properties [31].

Interestingly, the mechanism of neuropathic pain involves activation of the NMDA receptor (leading to the loss of downward inhibition) and inflammation-associated changes in the spinal cord [32,33,34,35,36]. Ketamine’s antagonism of the NMDA receptor meant that it was originally used as a painkiller.

S(+)-ketamine can also be used as antidepressant, although the NMDA receptor’s role in this depression has not been characterized [37]. S(+)-ketamine’s antidepressant effect is thought to come from activation of the mammalian target of rapamycin pathway and inactivation of glycogen synthase kinase-3 beta—especially because inhibitors of the latter enzyme potentiate and prolong the effects of ketamine, even at low doses [38,39].

### 2.3. Pharmacokinetics

Little is known about the pharmacokinetics of ACH derivatives. However, it is suspected that these derivatives are pharmacokinetically similar to ketamine, the only molecule used therapeutically, and for which detailed studies have been undertaken. However, sometimes very small structural modifications can greatly modify a drug’s pharmacokinetics. Given this context, we primarily describe the pharmacokinetics of ketamine below.

#### 2.3.1. Absorption

The bioavailability of orally administered ketamine is very low (between 8% and 24%), due to a major hepatic first pass effect. The bioavailability of orally administered S(+)-ketamine is even lower (~10%) [40,41,42]. The peak blood concentration is usually obtained after 40 to 55 min, and the first metabolites are detected after 10 to 30 min [6,42,43]. Similarly, the bioavailability of sublingual preparations (used for analgesic purposes) is low (~30%) [44].

In contrast, snorting or inhaling (the most frequent techniques in recreational use) leads to almost complete absorption [45]. The intramuscular route also allows rapid, high absorption, with a bioavailability of about 93%, and the injected ketamine can be detected in plasma after just 4 min [45,46]. After intravenous injection, ketamine can be observed after 5 min, and the plasma peak is seen at 5 to 30 min [45,47].

#### 2.3.2. Distribution

The ACH derivatives’ high lipid solubility (estimated by LogP) is an important factor for bioavailability and enables them to cross the various lipid membranes in tissues. The plasma protein-bound fraction of ketamine is low (around 10–30%), which allows the drug to cross the blood–brain barrier easily and, thus, to exert its best-known effects (anesthesia, analgesia and anti-inflammatory properties) in the brain [48].

After systemic absorption, ketamine (LogP = 2.2) distributes rapidly through the tissues, and especially the brain. Ketamine’s very short absorption half-life (2 to 4 min) gives it very rapid effects [6,48]. The drug’s volume of distribution is high (3.5 L/kg) due to its liposolubility [41,48,49] (Table 1).

#### 2.3.3. Metabolism

The metabolism of ketamine has been extensively studied, and is summarized in Figure 3 [26,50,51,52,53,54,55,56,57]. The primary metabolic pathway gives rise to norketamine (an active metabolite) and hydroxylated derivatives [45,46]. Most of these compounds are then glucuronoconjugated and excreted via the renal or biliary route. Ketamine is mainly metabolized in the liver by cytochromes P450 (CYP450); more particularly, CYP3A4 and CYP2B6 give rise to norketamine and dehydronorketamine (the other main metabolite), respectively [26,57] (Figure 3).

**Table 1 ijms-23-15574-t001:** The metabolic fate of ACH derivatives.

Molecule	Main Metabolites	Biotransformation	CYP Involved	Predicted LogP	Reference
	**Ketamine derivatives**				
DCK	Dihydro-DCK	Hydrogenation	Unknown	2.7	[58]
	Dihydro-nor-DCK	Hydrogenation + demethylation			
2F-DCK	Nor-2F-DCK	Demethylation	Unknown	2.9	[21,58,59]
	Dihydro-2F-DCK	Hydrogenation			
	Dihydro-nor-2F-DCK	Hydrogenation + demethylation			
	**Phencyclidine derivatives**				
PCP	c-PPC		CYP1A,3A	3.6	[60,61]
	t-PPC	Hydroxylation			
	PCHP				
3-OH-PCP	M1	Hydroxylation	Unknown	3.3	[62]
	M2	N-dealkylation + carboxylation			
	M3	O-glucuronidation			
3-MeO-PCP	Hydroxy-3-MeO-PCP	Hydroxylation	CYP2B6	3.6	[63,64,65]
	Demethyl-dihydroxy-3-MeO-PCP	Demethylation + hydroxylation	CYP2C19/2D6		
	Piperidine-dihydroxy-3-MeO-PCP	Hydroxylation	CYP2B6		
	**Eticyclidine derivatives**				
Methoxpropamine	N-despropyl(nor)MXPr	Depropylation	Unknown	2.8	[58,66]
	O-desmethylMXPr	Demethylation			
	DihydroMXPr	Hydrogenation			
2-oxo-PCE	2-en-PCA-N-Glu	Dehydration + glucuronidation	Unknown	2.5	[67]
	M3	Oxidative deamination + dehydration			
	O-PCA-N-Glu	Glucuronidation			
2-FDCNEK	2-fluorodeschloro-norketamine	N-dealkylation	Unknown		[68]

LogP was calculated with XLogP3 software. DCK: Deschloroketamine, PCP: Phencyclidine, MXPr: Methoxpropramine, PCE: Eticyclidine; c-PPC: cis-1-(1-phenyl-4-hydroxycyclohexyl)piperidine; t-PPC: trans-1-(1-phenyl-4-hydroxycyclohexyl)piperidine; 1-(1-phenylcyclohexyl)-4-hydroxypiperidine; 2-FDCNEK: 2-fluoro-deschloro-N-ethyl-ketamine aM: metabolite.

#### 2.3.4. Excretion

Ketamine has a relatively short duration of action. The excretion half-life is approximately 2 to 4 h, and the drug is excreted mainly by the kidneys [40]. Only 2% of ketamine and norketamine are excreted in unchanged form; the corresponding percentage for dehydronorketamine is about 16% [35,50,65,66,67]. Approximately 80% of hydroxyketamine and hydroxynorketamine are glucuronoconjugated, which facilitates biliary and urinary excretion [69]. The excretion half-life of S(+)-ketamine appears to be longer (about 4 to 7 h) than that of the racemic mixture, due to CYP450′s greater stereoselectivity for R(–)-ketamine [40,42,43,70,71].

The limited data on the excretion of other ACH derivatives come from studies of human metabolism [21,59,63,72]. 2F- DCK and 3-MeO-PCE were detected in the urine and bile of an intoxicated patient, suggesting the presence of both urinary and biliary excretion. Similarly, 2-F-DCK metabolites (including phase II metabolites) were found in the urine and bile of an intoxicated patient who subsequently died [19].

## 3. Arylcyclohexylamines: Clinical Toxicity and Forensic Cases

### 3.1. Clinical Toxicity

Ketamine has long been neglected by recreational users—partially because its effects were considered to be too strong. Despite its low addictive power and mild withdrawal syndromes, ketamine remains a highly toxic drug when taken under poorly controlled conditions. Psychic dependence and craving phenomena have often been reported. Ketamine is also often taken in combination with other molecules, which makes the drug’s effects even harder to predict. Most of the fatal cases of intoxications involve the use of multiple substances.

Ketamine was originally used as an anesthetic, but the absence of a respiratory depressant effect has caused the drug’s role to change gradually over time [73,74]. Ketamine has mainly been used recreationally by experienced users who wanted to discover new sensations or new emotions. The consumption of ketamine can induce euphoric effects, such as “cottony” drunkenness, at low doses, and hallucinations and powerful dissociation effects at high doses.

The ketamine trafficking and dealing network is very small, and is currently limited to free parties and (in some large cities) clubs. Ketamine has a reputation for not being cut, which has improved its image among consumers. For recreational use, the ingested or snorted doses are typically between 125 and 500 mg [75].

#### 3.1.1. Acute Toxicity

Initially, ketamine was only used by a very small proportion of people in the “rave” culture, i.e., a population of nomads who used the drug for its powerful dissociative and hallucinogenic effects. Ketamine had a rather negative reputation, due to its radical effects and its use in veterinary medicine.

This poor reputation faded in the 2000s, when ketamine experienced a revival among recreational users who were younger, less extreme, more regular consumers of lower doses. These low doses produce a drowsy sensation but do not go as far as the so-called “K-hole”—a “black hole” effect characteristic of ketamine that can include cognitive and amnesic disorders, mood and behavioral problems, hallucinatory delirium, nightmares and loss of identity and contact with reality. This new form of consumption is more conventional, and ketamine is now used more frequently in nightclubs and among friends (“ketamine aperitifs”). A group of inexperienced users consumes ketamine in a more unreasonable way (often combined with high doses of alcohol), and actively seeks the “K-hole” and loss of consciousness.

The main mode of use is snorting, although some people inject ketamine intravascularly or intramuscularly.

Few cases of death by overdose have been described, notably because chronic users and poly-users tend to be aware of the doses that should not be exceeded [76,77,78,79]. The main injuries reported (bruising, fractures, drowning, etc.) are related to the drug’s anesthetic effect.

In France, ketamine is relatively inexpensive, at between 40 and 50 euros per g; this quantity can be divided into several doses [80].

Users also like the fact that in contrast to alcohol, police forces cannot perform roadside tests for ACHs. Furthermore, ketamine’s effects wear off quickly (after 20 to 40 min) and so users can (for example) drive a motor vehicle soon after taking the drug.

In addition to the marked neurological effects, digestive disorders (such as diarrhea, nausea and vomiting), tachycardia and hypertension have also been described [81,82,83].

The medical management of these patients is sometimes complicated by the fact that several other substances have been ingested concomitantly. Management is essentially based on symptomatic treatment, with hyperhydration (to promote substance excretion) and tracheal intubation (to counter the effects of sedation). Clinical biochemistry variables (especially cytolysis markers, creatinine and creatine phosphokinase) and vital signs (arterial oxygen saturation, heart rate and blood pressure) must be monitored closely. An antipsychotic may also need to be administered, as the patients are often agitated [84].

#### 3.1.2. Chronic Toxicity

The toxicity profile of chronic intake is still unclear, but appears to include severe, irreversible damage to the bladder, such as thickening of the bladder wall, leading to secondary kidney damage [85]. Although there are no data on the bladder toxicity of other ACH derivatives, specific clinical monitoring for this entire chemical family makes sense. Digestive disorders and abdominal pain (known as “K-pain”, ranging from simple abdominal pain to intense colic) have also been observed [73,86]. Liver damage has also been described: in 2017, 10 cases of choledochal cysts and dilatation of the bile duct were linked to repeated and/or prolonged use of high-dose ketamine (>100 mg/d). Four of these cases required liver transplantation [73,86,87,88,89]. Cognitive problems include addictive behavior, impaired color perception, loss of memory and attention, longer reaction times, impaired perception of time and dissociative effects [90,91].

Ketamine tolerance in humans has been described, although there have been few cases of withdrawal syndrome. Cases of dependence have also been described [88,92,93,94,95]. The arrival of new synthetic derivatives on the market will doubtless change this safety profile, with greater variability in doses and product quality.

### 3.2. Epidemiology

The epidemiology of users of ketamine and its derivatives has changed markedly since the 2000s, with the appearance of a new wave of younger people using lower doses, but in a chronic manner [80]. Ketamine is usually supplied as a liquid formulation for injection, but can be prepared as small white crystals, or even (to facilitate use) as a powder. Some sellers report that ketamine is bought in 1- to 10-liter drums in India, sometimes colored, and then transported in shampoo bottles or other smaller containers. The drug can then be “cooked” into a solid form for easier use. A few cases of theft of hospital or veterinary supplies of ketamine (often of higher quality) have been reported [96]. The drug’s nicknames include “keta”, “ké”, “kéké”, “special K”, “kate”, “pony drug”, and “horse drug”. With the advent of the Internet and the Dark Web, many novel ACH derivatives have attracted interest. Polyconsumption is very frequently described, because ketamine is known to be easily combined with other molecules (e.g., cocaine) and helps to moderate the withdrawal symptoms of other drugs. Asia, and especially Hong Kong, have been greatly affected by the arrival of novel ACH derivatives since the 2000s [18,97]. Even though the raw materials are not easy to obtain, novel ACH derivatives now seem to be spreading across over the world, and consumption is growing exponentially. Ketamine consumption is lower in Europe than in Asia, although the United Kingdom and Spain are markets hubs for drugs produced in Asia. A European Monitoring Center for Drugs and Drug Addictions (EMCDDA) report estimated that only 16 ACHs were reported for the first time between 2005 and 2017—far fewer than the novel synthetic cathinones and cannabinoids [98]. The French TREND/SINTES network has reported significant increases in ketamine dealing and use. Supplies in France are, nevertheless, very limited for the moment, with most of the drug coming from India, the United Kingdom and Spain [99]. Table 2 summarizes the reported cases of ACH-derivative intoxication worldwide, and Figure 4 shows the cases’ geographical distribution.

#### 3.2.1. Phencyclidine Derivatives


**
*Phencyclidine*
**


In 2013, the US Center for Behavioral Health Statistics and Quality reported that PCP use and related emergency department admissions had increased substantially since the 2000s. Consumption increased by 400% between 2005 and 2011, and then by 200% between 2009 and 2011. This report, as well as Domici et al.’s (2015) observations in a Philadelphia hospital, show that consumption fluctuates greatly over time [121,122].


**
*4-MeO-PCP*
**


Although 4-MeO-PCP was synthesized in the 1960s, it appeared on the narcotics market in 2008 as one of the first “dissociative research chemicals” [4]. Many isomers or other structurally similar molecules have since appeared, which circumvent the legislation on drugs of abuse.

The first notification of 4-MeO-PCP from the European authorities dates back to 2012 [123]. Users described oral intakes between 50 and 100 mg, and 4-MeO-PCP appeared to be less potent than PCP and 3-MeO-PCP. Few cases of intoxication have been described, other than a case in Sweden in 2015 and a case in Korea in 2019 [17,124] (Table 2).


**
*3-MeO-PCP*
**


3-MeO-PCP is an isomer of 4-MeO-PCP. Few metabolic and kinetic data are available, although several cases of intoxication have been published [20,63,107]. 3-MeO-PCP was first noted by the EMCDDA in 2012 [123]. This molecule has been reported in two studies performed in the toxicology laboratory of the Rennes University Medical Center (Rennes, France). Berar et al. (2019) first described 3-MeO-PCP intoxication in a patient, with initial toxic concentrations in blood (71.1 ng/mL) and urine (706.9 ng/mL) [100]. Allard et al. (2019) then used molecular networking to describe the hepatic metabolism of 3-MeO-PCP [63]. In another French study, Grossenbacher et al. (2018) described five cases of 3-MeO-PCP intoxication (two of which were fatal) [108] (Table 2).


**
*3-OH-PCP*
**


3-OH-PCP was first reported in 2009. The drug is reportedly active at low doses (between 1 and 10 mg). However, no cases of intoxication have yet been described in the literature.

#### 3.2.2. Ketamine Derivatives

The “DCKs” (deschloroketamine (DCK) and the fluorinated derivative 2F-deschloroketamine (2F-DCK)) are the least well-described ACH derivatives. The first report on these molecules dates to 2015, with the analysis of powders found in the USA, China and Europe [125]. DCK and 2F-DCK are mainly available on the Dark Web [126]. These derivatives are particularly consumed in Asia in general, and in Hong Kong in particular. Although DCK was synthesized by Stevens in 1962, its misuse was not reported until 2015, and the first analytical data were published in 2016 (187,188).

2F-DCK was first synthesized in 1987, and consumption was first described in Internet forums in 2015 [59]. Between January and July 2019, Tang et al. described 20 cases of patients in whom 2F-DCK was detected [59]. In most cases, 2F-DCK was combined with ketamine. Tang et al. reported that 2F-DCK had the same toxic effects as ketamine, although poly-use prevented the identification of specific clinical effects [127]. Similarly, Weng et al. (2020) described 11 cases of 2F-DCK or DCK poisoning [128]. To broaden the detection window, Davidsen et al. (2020) developed a hair analysis technique, which provided the first pharmacokinetic data from samples collected from roadside offenders in Denmark [129]. Snorted doses of 2F-DCK are generally between 87.5 and 330 mg, whereas those for ketamine are between 60 and 250 mg [75]. There are only two case reports on 2F-DCK consumption: a fatal case of DCK and 2-oxo-PCE intoxication in Germany [113] and our report on a patient at Lille University Medical Center (Lille, France) [19] (Table 2).

#### 3.2.3. Eticyclidine Derivatives

Like the other molecules described here, eticyclidine derivatives are particularly present in Asian countries, and especially in Hong Kong. Although these derivatives are not very common, a few cases of intoxication have yielded clinical and toxicological data (Table 2).


**
*Methoxetamine*
**


Methoxetamine (3-MeO-2-oxo-PCE) appears to be the most widespread ACH in Europe, as dealers sometimes supply it instead of ketamine when supplies of the latter run out [9,17,100,103,130]. Its effects are reportedly similar to those of ketamine, but last longer and are more intense [81,131,132]. The doses are also very different, meaning that the switch from ketamine is particularly dangerous. The first consumer discussions on Internet forums date back to 2010 [133]. Methoxetamine’s nicknames include “Mexxy”, “M-ket”, “MEX”, “Kmax”, and “legal ketamine” [134]. It is the most widely used ketamine derivative in Europe, prompting the European authorities to classify it as a narcotic in 2013 [135]. This is also the case in Japan, the United Kingdom, China, Russia, Turkey and Korea [136]. Nasal and oral administration routes are the most common, although cases of intravenous, anal and sublingual administration have been described [132,134,137]. The most commonly reported doses are 20 to 100 mg for the nasal route, 40 to 100 mg for the oral route and 10 to 80 mg for the intramuscular route [131]. The first effects are felt after 10 to 90 min, with a peak between 1 and 7 h (depending on the administration route) [132,137]. In 2015, the World Health Organization reported 120 cases of non-fatal poisoning and 22 deaths worldwide [136]. The adverse effects are similar to those of ketamine, with nausea/diarrhea/vomiting, tachycardia, hypertension and loss of consciousness [81,82,83] (Table 2).


**
*2-oxo-PCE*
**


The first report to EMCDDA was made in 2016 [98]. Tang et al. (2018) described a cluster of 56 cases of intoxication in the Hong Kong area in October 2017. The toxidrome was similar to that of ketamine, with predominant tachycardia and hypertension. Users report dissociative and hallucinogenic effects similar to those of ketamine, but more powerful (five times more than ketamine and three times more than methoxetamine). Oral doses are reportedly between 6 and 12 mg [138,139]. Nasal swab samples from three users contained only 2-oxo-PCE and no ketamine [97] (Table 2).


**
*3-MeO-PCE*
**


Although an assay for 3-MeO-PCE in biological fluids has been developed and validated, there are no published in vivo data on concentrations or metabolism [140].

Chemists have recently developed other structurally similar PCP derivatives (e.g., diarylethylamines), but these are not yet widely used [141] (Table 2).

As reported in Table 2, ACHs are frequently co-consumed with other drugs of abuse, including alcohol, THC, amphetamines or cocaine. These cocktails are likely to modify the pharmacokinetics, pharmacodynamics and toxicity profile of these drugs.

## 4. Conclusions

ACH consumption is growing exponentially, and is associated with an increase in the number of cases of fatal intoxication. The rapid detection of ACHs is a challenge for toxicology laboratories. Given the small number of reports on intoxication, few data on the metabolism and clinical effects of ACHs have been published. The number of cases has certainly been underestimated because the molecules are not detected or not easily detected. However, constant improvement in analytical techniques and greater access to databases should enable the epidemiology of this phenomenon to be better defined in the future. Assessment of the ACHs’ metabolism and pharmacologic properties might make it possible to provide more suitable medical management, rather than symptomatic treatment alone. Metabolic studies will provide insights into the fate of these drugs in the body, the presence of active metabolites, the metabolic pathways involved and, potentially, valuable biomarkers for consolidating diagnostic evidence and broadening detection windows when the parent molecule is no longer detectable.

## Figures and Tables

**Figure 1 ijms-23-15574-f001:**
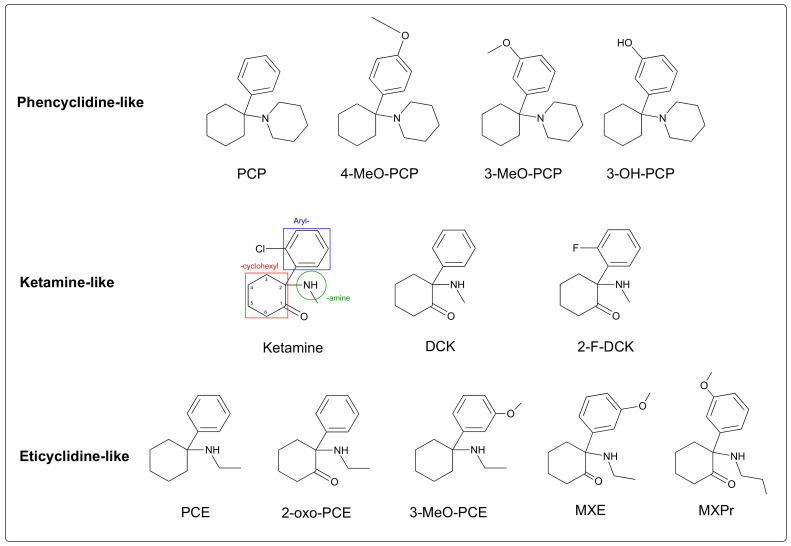
The ACH derivatives most frequently involved in cases of intoxication. PCP: phencyclidine (1-(1-phenylcyclohexyl)piperidine); DCK: deschlorokétamine (2-(methylamino)-2-phenylcyclohexan-1-one); PCE: eticyclidine (N-ethyl-1-phenylcyclohexan-1-amine); MXE: methoxetamine (2-(ethylamino)-2-(3-methoxyphenyl)cyclohexan-1-one); MXPr: methoxpropamine (2-(3-methoxyphenyl)-2-(propylamino)cyclohexan-1-one).

**Figure 2 ijms-23-15574-f002:**
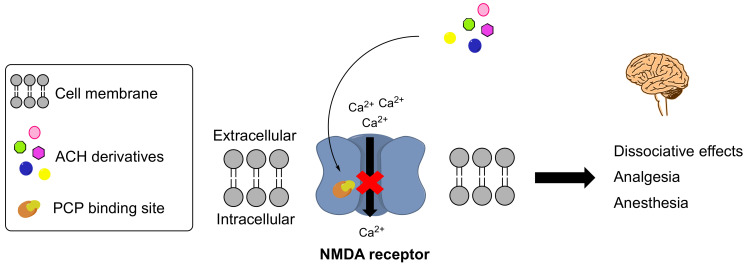
Mechanism of action of arylcyclohexylamine derivatives.

**Figure 3 ijms-23-15574-f003:**
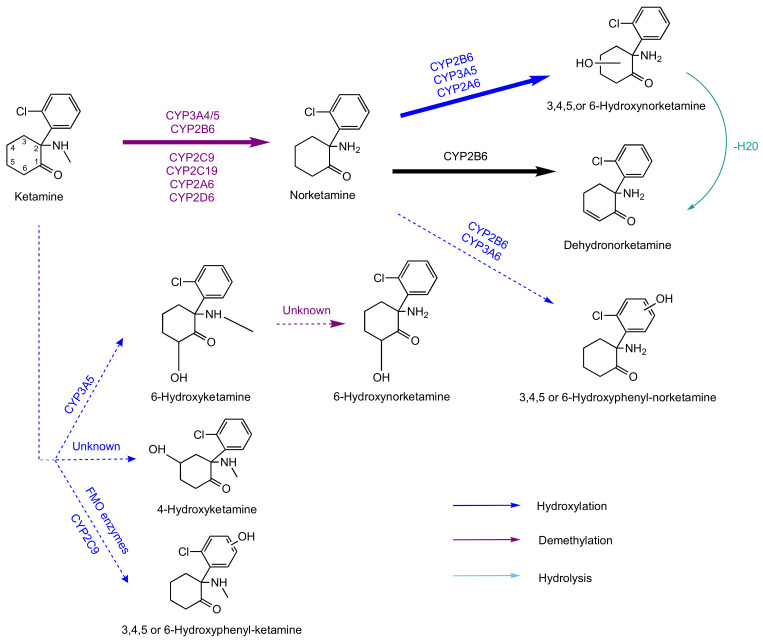
Overview of ketamine’s metabolism [26,45,46,51,52,53]; bold arrows indicate the major pathway and dotted arrows indicate minor metabolic pathways; FMO: flavin-containing monooxygenase.

**Figure 4 ijms-23-15574-f004:**
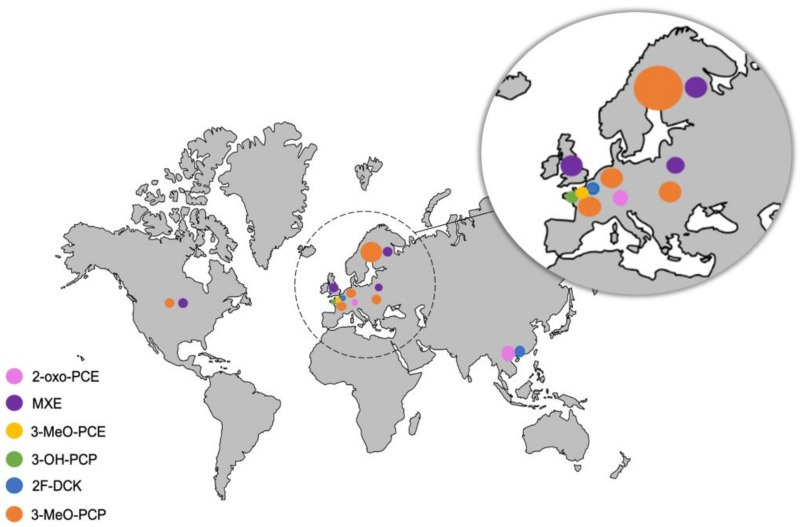
Geographical distribution of published cases of ACH-derivative intoxication.

**Table 2 ijms-23-15574-t002:** Literature reports of intoxications involving ACH derivatives other than ketamine.

	Co-Intoxication	Matrix	Concentration	Mortality	Country	Source
**3-MeO-PCP**	No	Peripheral blood, urine	Case 1: Peripheral blood = 71.1 ng/mL and urine = 706.9 ng/mL/Case 2 none	2 non-fatal	France	[100]
	Ethanol, diazepam, cocaine	Femoral blood, urine, bile, hair	Femoral blood = 63 ng/mL, bile = 64 ng/mL, urine = 94 ng/mL	1 fatal	France	[101]
	Methadone, THC	Femoral blood, urine	Femoral blood = 3 525 ng/mL and urine = 7 384 ng/mL	1 fatal	France	[64]
	Alcohol	Peripheral blood, urine	Case 1: Peripheral blood = 350.0 ng/mL and urine = 6109.2 ng/mL/Case 2: peripheral blood = 180.1ng/mL and urine = 3003.5 ng/mL	2 non-fatal	Italy	[102]
	4 cases with 4-MeO-PCP, but the majority included other novel derivatives	Serum, urine	Serum = 1 tp 242 ng/mL, urine = 2 to 52,759 ng/mL	56 cases	Sweden	[17]
	Several substances	Femoral blood	Serum = 0,05–0,38 μg/g	1 fatal and 7 non-fatal	Sweden	[103]
	Diphenydramine, THC, amphetamine	Postmortem blood	Blood = 139 ± 41 µg/L	1 fatal	USA	[104]
	Amphetamine, alcohol	Postmortem blood	Blood = 152 µg/L	1 fatal	The Netherlands	[105]
	Methamphetamine (Case 1), Ethanol/bupropion/paroxetine (Case 2)	Postmortem blood	Blood = Case 1: 0.63 and Case 2: 3.2 mg/L	2 fatal	USA	[106]
	3-OH-PCP, 3-MeO-PCP, 2F-DCK, N-ethylhexedrone, CMC	Urine	Urine = 110 mg/L	1 non-fatal	France	[72]
	Alcohol, amphetamine (Case 1)	Peripheral blood, urine	Blood = 49 ng/mL (Case 1) and 66 ng/mL (Case 2)	2 non-fatal	Czech Republic	[107]
	Several substances	Peripheral and femoral blood, urine, bile, hair	Urine = 498 ng/mL to 16,700 ng/mL Blood = 63 ng/mL	3 non-fatal and 2 fatal	France	[108]
	No data	Femoral blood, urine	Case 1: Femoral blood = 63 ng/mL, bile = 94 ng/mL/Case 2: femoral blood = 498 ng/mL and urine = 16 700 ng/mL	2 fatal	France	[20]
	No data	Urine	Qualitative test	1 non-fatal	USA	[109]
**2F-DCK**	3-MeO-PCE, 5-MeO-DMT, amphetamine and cocaine	Peripheral blood, urine, bile, vitreous humor	Peripheral blood = 1780 μg/L, urine = 6106 μg/L, bile = 12,200 μg/L, vitreous humor =1500 μg/L	1 fatal	France	[19]
	Majority of ketamine and derivates, cocaine, methamphetamine	Urine	Qualitative test	20 cases	Hong Kong	[59]
	3-MeO-PCP, 3-OH-PCP, CMC, N-ethylhexedrone	Urine	Urine = 147 mg/L	1 non-fatal	France	[72]
**3-OH-PCP**	3-MeO-PCP, CMC, 2F-DCK, N-ethylhexedrone	Urine	Urine = 12,085 mg/L	1 non-fatal	France	[72]
**3-MeO-PCE**	2F-DCK, 5-MeO-DMT, amphetamine and cocaine	Peripheral blood, urine, bile, vitreous humor	Peripheral blood = 90 μg/L, urine = 6.3 µg/L, bile = 3.5 μg/L, vitreous humor = 66 μg/L	1 fatal	France	[19]
**MXE**	Several substances	Peripheral blood, urine	Qualitative test	8 fatal	UK	[110]
	Alcohol/benzofuran (Case 1)	Serum	Serum = 0.09 to 0,2 mg/L	3 non-fatal	UK	[83]
	Methamphetamine, dextromethorphan	Serum	Serum = 160 ng/mL	1 non-fatal	USA	[111]
	Three synthetic cannabinoids	Femoral blood	Femoral blood = 8.6 µg/g	1 fatal	Sweden	[112]
	Clonazepam, THC, diphenhydramine, MDMA	Peripheral blood	Blood = 10 ng/mL	1 fatal	USA	[113]
	Amphetamine	Peripheral blood	Peripheral blood = 0.32 μg/ml	1 fatal	Poland	[114]
	AH-7921, benzodiazepines	Serum	Unknown	1 non-fatal case	Norway	[115]
	5- or 6-APB	Serum	Serum = 0.16 to 0.45 mg/L	3 non-fatal case	United Kingdom	[116]
	Ketamine, psychotics	Serum, urine, hair	Serum = 30 and urine = 408 µg/L	1 non-fatal case	France	[117]
	No data	Serum, urine	Serum = 5.8 μg/mL, urine = 85 μg/mL	1 fatal	Poland	[118]
	No data	Serum, urine	Serum = 270 ng/ml and urine = 660 ng/mL	1 non-fatal	Poland	[119]
**2-oxo-PCE**	Several substances	Serum, urine	No data	56 non-fatal cases	Hong Kong	[97]
	Venlafaxine	Liver, urine, bile, gastric content, heart blood, femoral blood	Liver = 6137 ng/g, urine = 3468 μg/L, bile fluid = 3290 μg/L, gastric contents = 3086 μg/L, heart blood = 2159 μg/L liquor = 1564 μg/L, femoral blood = 375 μg/L	1 fatal case	Germany	[120]
**2-FDCNEK**	No data	Urine	Urine (metabolite only: 2-FDCNEK) = 133 ng/mL	1 non-fatal case	China	[68]

THC: tetrahydrocannabinol; CMC: chloromethcathinone; MDMA: 3,4-methyl-enedioxy-methamphetamine; ABP: 6-(2-aminopropyl)benzofuran); AH-7921: 3,4-dichloro-N-[[1-(dimethylamino)cyclohexyl]methyl]benzamide; 5-MeO-DMT: O-methyl-bufotenin.

## Data Availability

Not applicable.

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
