# Peer review of "Arylcyclohexylamine Derivatives: Pharmacokinetic, Pharmacodynamic, Clinical and Forensic Aspects"

_ijms, 2022, doi:10.3390/ijms232415574_

Round 1
Reviewer 1 Report
Review, entitled “Arylcyclohexylamine derivatives: pharmacokinetic, pharmacodynamic, clinical and forensic aspects” by Romain Pelletier et al. concerns pharmacokinetics, pharmacodynamics, epidemiology, chemistry, metabolism and signs of intoxication of ACH derivatives.
The review is well written and structured, but does not feet aims and scope of IJMS. I would recommend resubmitting this review to a journal with a more medical focus (for example, Medicina by MDPI).
Minor points:
1. Table 1: The table name must be capitalized. Abbreviations should be moved below the table.
2. Table 2: Abbreviations should be moved below the table.
3. The mention of Table 2 in the text should be written not in Roman, but in Arabic numerals (lines 293, 349, 354, 381, 387).
Author Response
Rennes, 10th November 2022
Editorial
International Journal of Molecular Sciences
Dear editor,
Please find enclosed our revised manuscript entitled: “Arylcyclohexylamine derivatives: pharmacokinetic, pharmacodynamic, clinical and forensic aspects” as a submission for publication in International Journal of Molecular Sciences.
We would like to thank you for your helpful and constructive feedback concerning our manuscript. We discussed all the comments and - where appropriate - amended the manuscript accordingly as indicated. For better clarification, please find enclosed a reply letter that addresses all of the comments. In the manuscript, we highlighted all changes in yellow.
This work has not been nor will be submitted for publication in any other journal until you have taken your final decision.
We hope that you will find it suitable for publication.
Thank you for your cooperation.
Sincerely yours.
Dr Romain Pelletier
REVIEWER #1:
Review, entitled “Arylcyclohexylamine derivatives: pharmacokinetic, pharmacodynamic, clinical and forensic aspects” by Romain Pelletier et al. concerns pharmacokinetics, pharmacodynamics, epidemiology, chemistry, metabolism and signs of intoxication of ACH derivatives.
The review is well written and structured, but does not feet aims and scope of IJMS. I would recommend resubmitting this review to a journal with a more medical focus (for example, Medicina by MDPI).
Minor points:
- Table 1: The table name must be capitalized. Abbreviations should be moved below the table.
- Table 2: Abbreviations should be moved below the table.
- The mention of Table 2 in the text should be written not in Roman, but in Arabic numerals (lines 293, 349, 354, 381, 387).
Response to reviewer:
Thank you for your revision, we correct all points that you mentioned.

Reviewer 2 Report
The aim of the presented review was to describe different features of arylcyclohexylamine (ACH) derivatives such as pharmacokinetic, pharmacodynamic, clinical and forensic aspects. It is interesting subject, review brings developed knowledge about this group of compounds especially important as such compounds are considered as new psychoactive substances. Three main groups of compounds were discussed: phencyclidine-, ketamine-, and eticyclidine-like structures. Their main metabolites were mentioned, toxicity of the metabolites and types of reactions leading to their construction under influence of several CYP isoforms and other enzymes. Clinical toxicity and forensic cases were reported. Epidemiology of users was inquired. It is very valuable scientific contribution because of growing number of new psychoactive substances at the market with unknown biological properties, not evaluated This manuscript fits into IJMS scope and could be published in IJMS, however it is written somehow chaotic, needs several changes before publication.
1) Figure 1:
- add structure PCE similarly as for other main representants of ACH compounds:
- name 3-MeO-PCE does not fit to structure drawn in Fig 1;
- MXE- does this compound contain carbonyl group?;
- MXPr has not propyl group
- To avoid mistakes add chemical names of compounds after giving their abbreviations
2) Figure 3:
- Check the routes of metabolism and structures of metabolites obtained under influention of different CYP isoforms
- Indicated with green arrow elimination of H2O does not occur from norketamine to dehydrnorketamine. Such product can be formed from 5- or 6- hydroxynorketmine
3) In Table 1 check the biotransformation of methoxpropamine (is it really demethylation: which kind O- or N-?; N – is impossible)
- Metabolic pathways for 2-oxo-PCE were not described
4) Table 2 (once quoted Table II (e.g. line 293) – it should be the additional paragraph added, problem of co-intoxication (almost not presented) should be more discussed; how were the discussed compounds chosen, which co-intoxicating compounds existed, which were the most dangerous effects of co-administration, etc
5) Add Table 1 caption- e.g. what means c-PPC,t-PPC, M1, M2, M3, PCHP
6) Add table 2 caption e.g. what means AH-7921, 5-MEO-DMT
7) Minor changes, reformulate sentences lines 205/206; 254
Author Response
Rennes, 10th November 2022
Editorial
International Journal of Molecular Sciences
Dear editor,
Please find enclosed our revised manuscript entitled: “Arylcyclohexylamine derivatives: pharmacokinetic, pharmacodynamic, clinical and forensic aspects” as a submission for publication in International Journal of Molecular Sciences.
We would like to thank you for your helpful and constructive feedback concerning our manuscript. We discussed all the comments and - where appropriate - amended the manuscript accordingly as indicated. For better clarification, please find enclosed a reply letter that addresses all of the comments. In the manuscript, we highlighted all changes in yellow.
This work has not been nor will be submitted for publication in any other journal until you have taken your final decision.
We hope that you will find it suitable for publication.
Thank you for your cooperation.
Sincerely yours.
Dr Romain Pelletier
REVIEWER #2:
The aim of the presented review was to describe different features of arylcyclohexylamine (ACH) derivatives such as pharmacokinetic, pharmacodynamic, clinical and forensic aspects. It is interesting subject; review brings developed knowledge about this group of compounds especially important as such compounds are considered as new psychoactive substances. Three main groups of compounds were discussed: phencyclidine-, ketamine-, and eticyclidine-like structures. Their main metabolites were mentioned, toxicity of the metabolites and types of reactions leading to their construction under influence of several CYP isoforms and other enzymes. Clinical toxicity and forensic cases were reported. Epidemiology of users was inquired. It is very valuable scientific contribution because of growing number of new psychoactive substances at the market with unknown biological properties, not evaluated. This manuscript fits into IJMS scope and could be published in IJMS, however it is written somehow chaotic, needs several changes before publication.
1) Figure 1:
- add structure PCE similarly as for other main representants of ACH compounds:
- name 3-MeO-PCE does not fit to structure drawn in Fig 1;
- MXE- does this compound contain carbonyl group?;
- MXPr has not propyl group
- To avoid mistakes add chemical names of compounds after giving their abbreviations
Response to Reviewer:
Thank you for your comments. We have corrected the chemical structure errors in the figure with the IUPAC name of each molecule. We have also added these IUPAC names after the abbreviations in the legend of Figure 1.
Figure 1. The ACH derivatives most frequently involved in cases of intoxication. PCP: phencyclidine (1-(1-phenylcyclohexyl)piperidine) ; DCK: deschlorokétamine (2-(methylamino)-2-phenylcyclohexan-1-one) ; PCE: eticyclidine (N-ethyl-1-phenylcyclohexan-1-amine) ; MXE: methoxetamine (2-(ethylamino)-2-(3-methoxyphenyl)cyclohexan-1-one) ; MXPr: methoxpropamine (2-(3-methoxyphenyl)-2-(propylamino)cyclohexan-1-one).
2) Figure 3:
- Check the routes of metabolism and structures of metabolites obtained under influention of different CYP isoforms
- Indicated with green arrow elimination of H2O does not occur from norketamine to dehydrnorketamine. Such product can be formed from 5- or 6- hydroxynorketmine
Response to reviewer:
Thank you for this remark, we have modified the figure accordingly. We also decided to reduce the number of references concerning this figure, taking into account only those which seemed to us the most relevant in the realization of this one.
3) In Table 1 check the biotransformation of methoxpropamine (is it really demethylation: which kind O- or N-?; N – is impossible)
Response to reviewer:
Thank you for your comment it is indeed a depropylation and not a demethylation. We corrected this point in Table 1.
- Metabolic pathways for 2-oxo-PCE were not described
Response to reviewer:
Metabolism pathway is not described for 2-oxo-PCE in the literature. We mentioned “unknown” in Table 1 when metabolism pathways are not described.
4) Table 2 (once quoted Table II (e.g. line 293) – it should be the additional paragraph added, problem of co-intoxication (almost not presented) should be more discussed; how were the discussed compounds chosen, which co-intoxicating compounds existed, which were the most dangerous effects of co-administration, etc
Response to reviewer:
It would indeed be very interesting to be able to study the effects of a co-consumption of these products but bibliographic data is scarce. Consequently, we added a paragraph following this table to highlight this issue.
“As reported in Table 2, ACH are frequently co-consumed with other drugs of abuse, including alcohol, THC, amphetamine or cocaine. These cocktails are likely to modify the pharmacokinetics, pharmacodynamics and toxicity profile of these drugs.”
5) Add Table 1 caption- e.g. what means c-PPC,t-PPC, M1, M2, M3, PCHP
Response to reviewer:
Thank you for this remark. We added this legend under Table 1.
6) Add table 2 caption e.g. what means AH-7921, 5-MEO-DMT
Response to reviewer:
Thank you for this remark. We added this legend under Table 2.
7) Minor changes, reformulate sentences lines 205/206; 254
“Most of the fatal cases intoxications involving the use of multiple substances.”
“The toxicity profile of chronic intake is still remains unclear but appears to include severe, irreversible damage to the bladder; in particular, thickening of the bladder wall can lead to secondary kidney damage (85).”
Response to reviewer:
We changed these sentences.
- “Most of the fatal cases intoxications involve the use of multiple substances.”
- “The toxicity profile of chronic intake is still unclear but appears to include severe, irreversible damage to the bladder, such as thickening of the bladder wall leading to secondary kidney damage (85).”

Round 2
Reviewer 1 Report
The authors have made all the corrections, but I still believe that the article does not feet aims and scope of IJMS. Therefore, the decision on the possibility of publication will be at the discretion of the Editor.
Author Response
Rennes, 18th November 2022
Editorial
International Journal of Molecular Sciences
Dear editor,
Please find enclosed our revised manuscript entitled: “Arylcyclohexylamine derivatives: pharmacokinetic, pharmacodynamic, clinical and forensic aspects” as a submission for publication in International Journal of Molecular Sciences.
We would like to thank you for your helpful and constructive feedback concerning our manuscript. We discussed all the comments and - where appropriate - amended the manuscript accordingly as indicated. For better clarification, please find enclosed a reply letter that addresses all of the comments. In the manuscript, we highlighted all changes in yellow.
This work has not been nor will be submitted for publication in any other journal until you have taken your final decision.
We hope that you will find it suitable for publication.
Thank you for your cooperation.
Sincerely yours.
Dr Romain Pelletier
REVIEWER #1:
The authors have made all the corrections, but I still believe that the article does not feet aims and scope of IJMS. Therefore, the decision on the possibility of publication will be at the discretion of the Editor.
Response to reviewer:
Thanks for judging our article and giving us some advice on the manuscript. We think that this review is very much in the scope of IJMS, as for example Patocka et al. 2020 regarding cathinones or Pagano et al. 2022 with cannabinoids. However, we leave the choice of acceptance to the editor.

Reviewer 2 Report
After corrections manuscript is suitable for publication in IJMS
Author Response
Rennes, 18th November 2022
Editorial
International Journal of Molecular Sciences
Dear editor,
Please find enclosed our revised manuscript entitled: “Arylcyclohexylamine derivatives: pharmacokinetic, pharmacodynamic, clinical and forensic aspects” as a submission for publication in International Journal of Molecular Sciences.
We would like to thank you for your helpful and constructive feedback concerning our manuscript. We discussed all the comments and - where appropriate - amended the manuscript accordingly as indicated. For better clarification, please find enclosed a reply letter that addresses all of the comments. In the manuscript, we highlighted all changes in yellow.
This work has not been nor will be submitted for publication in any other journal until you have taken your final decision.
We hope that you will find it suitable for publication.
Thank you for your cooperation.
Sincerely yours.
Dr Romain Pelletier
REVIEWER #2:
After corrections manuscript is suitable for publication in IJMS.
Response to Reviewer:
Thank you for reviewing our manuscript and providing valuable comments.

Round 3
Reviewer 1 Report
If the editors consider that the review is a good fit for the journal, then it may be published in the IJMS.